# Association between Ready-to-Eat Cereal Consumption and Nutrient Intake, Nutritional Adequacy, and Diet Quality in Adults in the National Health and Nutrition Examination Survey 2015–2016

**DOI:** 10.3390/nu11122952

**Published:** 2019-12-04

**Authors:** Yong Zhu, Neha Jain, Vipra Vanage, Norton Holschuh, Anne Hermetet Agler, Jessica D. Smith

**Affiliations:** 1Bell Institute of Health and Nutrition, General Mills, Inc., Minneapolis, MN 55427, USA; aha28@cornell.edu (A.H.A.); jessica.smith@genmills.com (J.D.S.); 2Global Knowledge Solutions, General Mills India Pvt. Ltd., Mumbai, Maharashtra 400076, India; neha.jain@genmills.com (N.J.); vipra.vanage@genmills.com (V.V.); 3Global Knowledge Solutions, General Mills, Inc., Minneapolis, MN 55427, USA; nort.holschuh@genmills.com

**Keywords:** ready-to-eat cereal, dietary intake, national health and nutrition examination survey

## Abstract

This study examined differences in dietary intake between ready-to-eat cereal eaters and non-eaters in adults from the United States. Participants (*n* = 5163) from the National Health and Nutrition Examination Survey 2015–2016 were included. One-day dietary recall was used to define ready-to-eat cereal consumption status and estimate dietary intake in eaters and non-eaters. Data from Food Patterns Equivalent Database 2015–2016 were used to compare intakes of food groups by consumption status. Diet quality was assessed by Healthy Eating Index 2015. Nineteen percent of US adults were ready-to-eat cereal eaters; they had a similar level of energy intake as non-eaters, but they had significantly higher intake of dietary fiber, and several vitamins and minerals, such as calcium, iron, magnesium, potassium, zinc, vitamin A, thiamin, riboflavin, niacin, vitamin B_6_, folate, vitamin B_12_, and vitamin D. They were also more likely to meet nutrient recommendations. Compared to non-eaters, ready-to-eat cereal eaters had the same level of added sugar intake but they had significantly higher intake of whole grains, total fruits, and dairy products. The diet quality of ready-to-eat cereal eaters was significantly higher than that of non-eaters. The study supports that ready-to-eat cereal eaters have better dietary intake with a healthier dietary pattern than non-eaters in the United States.

## 1. Introduction

Ready-to-eat (RTE) cereal is a common breakfast food in the US. It has been consistently reported that higher intake of RTE cereal is associated with significantly higher intake of certain vitamins and minerals such as thiamin, riboflavin, niacin, folate, vitamin D, iron, and zinc among adults in the US [1,2,3,4,5] and other countries [6,7,8,9]. Nonetheless, most of the previous studies focused on RTE cereal consumed at breakfast alone and its association with daily dietary intake. A recent study indicated about 20% of RTE cereal was consumed at non-breakfast occasions in US adults [10]; therefore, examining RTE cereal consumed at both breakfast and non-breakfast occasions would further quantify nutrition contribution of RTE cereal consumption to daily dietary intake.

The 2015–2020 Dietary Guidelines for Americans recommends whole grain fortified RTE cereal to increase consumption of several under-consumed nutrients of public health concern [11]. However, RTE cereal has been scrutinized for its sugar content. Information about the current consumption status of RTE cereal and its association with dietary intake is critical as it may inform upcoming 2020–2025 Dietary Guidelines for Americans, nutrition policy, and product reformulation. A recent study found that US children who reported consuming RTE cereal had higher diet quality [12]. This study sought to determine if the same association between RTE cereal consumption and diet quality was present among US adults.

Therefore, the objective of this study was to compare nutrient intake, nutritional adequacy, and diet quality between RTE cereal eaters and non-eaters in the US adult population using the most recent nationally representative data. In addition, the prevalence of RTE cereal consumption over time from 2003 to 2016 was examined. It was hypothesized that RTE cereal consumption would be associated with increased intake of vitamins and minerals, as well as better diet quality and a higher likelihood of meeting nutrient intake recommendations.

## 2. Materials and Methods

### 2.1. Data Source and Population

The National Health and Nutrition Examination Survey (NHANES) is a nationally representative, cross-sectional survey that collects 24-h dietary recalls and health related data in the non-institutionalized US population; the data have been released in two-year cycles since 1999 and are publicly available. The NHANES study protocol was approved by the National Center for Health Statistics Ethics Review Board; adult participants provided their written informed consent. 

The NHANES 2015–2016 data [13] were used for the main analysis. Two 24-h dietary recalls were available in NHANES; the first dietary recall was conducted at a mobile examination center and the second dietary recall was conducted through a telephone interview 3–10 days later. The response rate for interviews in NHANES 2015–2016 was 61.3% [14]. Both dietary recalls were collected using the Automated Multiple Pass method [15,16] and the dietary intake data were linked to the United States Department of Agriculture (USDA) Food and Nutrient Database for Dietary Studies (FNDDS) 2015–2016 [17] to estimate nutrient intake from each reported food and beverage. FNDDS includes the nutritional composition of all foods and beverages reported in NHANES including both those prepared at home and purchased [18]. The USDA National Nutrient Database for Standard Reference Release 28 (SR 28) is used to determine the nutritional value of foods and beverages in FNDDS; approximately one-third of foods and beverages in FNDDS are a direct match to items in SR 28 and a recipe calculation approach is used for the remaining foods [18]. The Food Patterns Equivalents Database (FPED) 2015–2016 [19] data were linked to the NHANES dietary data to estimate consumption data of different food groups. 

Participants aged 18 years or older whose Day 1 dietary recall status was reliable, as determined by the NHANES, were included in this study (*n* = 5266). According to NHANES 2015–2016, a reliable dietary recall status was defined as an individual completing the first four steps of the five-step Automated Multiple Pass method and if foods/beverages consumed for each reported eating occasion were identified [13]. Of those, pregnant or lactating women were excluded (*n* = 103) due to their different nutrition needs. Therefore, the present study had a final sample size of 5163. Among these participants, adults aged 18–54 years (*n* = 3099) and adults aged 55 years or older (*n* = 2064) were analyzed separately. 

### 2.2. RTE Cereal Consumption

Similar to previous studies using NHANES data [20,21,22,23,24], Day 1 dietary data were used in this study, which provide an appropriate estimate of population mean intake of nutrients on a given day [25]. Participants were classified as RTE cereal eaters or non-eaters depending on whether they reported consumption of RTE cereal food in their Day 1 dietary recall. RTE cereal foods were classified using USDA’s What We Eat In America (WWEIA) classification system based on how they are typically consumed and their nutrient content. All foods identified by WWEIA as “ready-to-eat cereal, high sugar (>21.2 g/100 g)” and “ready-to-eat cereal, low sugar (≤21.2 g/100 g)” were used to define RTE cereal in this study regardless of its nutrient profiles or reported brand information. For descriptive purposes, Day 1 dietary data from NHANES 2003–2004 and thereafter were also used to calculate the percentage of RTE cereal consumers in US adults in each NHANES cycle, to understand RTE cereal consumption patterns over time, after the same inclusion and exclusion criteria listed above were applied.

### 2.3. Outcome Variables

#### 2.3.1. Macro- and Micro-Nutrients

Total energy intake and nutrient intake were obtained from total dietary intake data from Day 1. Intake from dietary supplements was not included as the 2015–2020 Dietary Guidelines for Americans recommends foods to be the primarily sources for nutritional needs [11]. The list of nutrients examined in this study included total carbohydrate, total sugars, added sugar, dietary fiber, total fat, saturated fat, protein, calcium, iron, magnesium, potassium, sodium, zinc, vitamin A, thiamin, riboflavin, niacin, vitamin B6, folate, vitamin B12, vitamin C, vitamin D, and vitamin E. Those nutrients were identified as under-consumed nutrients, or nutrients of public health concern, from the 2015–2020 Dietary Guidelines for Americans, or those typically fortified in RTE cereals. Percent difference in intake was calculated as: (mean intake in RTE cereal eaters − mean intake in non-eaters)/mean intake in non-eaters × 100%.

#### 2.3.2. Percent Contribution of RTE Cereal to Daily Nutrient Intake

To understand public health impact of RTE cereal consumption, contribution of nutrients from RTE cereal consumed to total nutrient intake in RTE cereal eaters, as well as in the total population, was calculated using Day 1 dietary data. Contribution of RTE cereal and co-consumed milk combined to daily nutrient intake was also estimated in RTE cereal eaters. 

#### 2.3.3. Percent of the Population Below Estimated Average Requirement 

Percentages below Estimated Average Requirement (EAR) for vitamin A, thiamin, riboflavin, niacin, vitamin B_6_, folate, vitamin B_12_, vitamin C, vitamin D, calcium, iron, and zinc were calculated using the National Cancer Institute Usual Intake Method based on both Day 1 and Day 2 24-h dietary data and adjusted for interview sequence and weekday of the recall [26]. Using two days of dietary recalls for usual intake accounts for within-subject variations in dietary intake; intake on a single day is not a robust estimate for usual intake for nutrition assessment such as comparisons with dietary reference intakes [27]. Standard errors for percentages below EAR were calculated using the Balanced Repeated Replication method [28].

#### 2.3.4. Food Group Intake

Food group intake in RTE cereal eaters and non-eaters was calculated using Day 1 FPED data, which disaggregate all foods according to standard recipes to calculate the intake of each food group in cup equivalents or ounce equivalents [19]. Intake of food groups including total dairy, total fruit, total vegetables, and total protein foods, as well as subgroups possibly associated with RTE cereal consumption, such as fluid milk, intact fruit, fruit juice, whole grains, and refined gains, were calculated. 

#### 2.3.5. Healthy Eating Index 2015 

Dietary quality was estimated using the Healthy Eating Index 2015 (HEI-2015) [29], which was a measure for compliance to the 2015–2020 Dietary Guidelines for Americans. The HEI-2015 score is based on the sum of 13 components for a maximum of 100 points (highest diet quality) [29]. Points are given for each component according to intake per 1000 kcal or percent of energy. The maximum and minimum scores for each component are as follows: total fruits (5 points: ≥0.8 cup eq./1000 kcal; 0 point: no intake), whole fruit (5 points: ≥0.4 cup eq./1000 kcal; 0 point: no intake), total vegetables (5 points: ≥1.1 cup eq./1000 kcal; 0 point: no intake), greens and beans (5 points: ≥0.2 cup eq./1000 kcal; 0 point: no intake), whole grains (10 points: ≥1.5 oz eq./1000 kcal; 0 point: no intake), dairy (10 points: ≥1.3 cup eq./1000 kcal; 0 point: no intake), total protein foods (5 points: ≥2.5 oz eq./1000 kcal; 0 point: no intake), seafood and plant proteins (5 points: ≥0.8 cup eq./1000 kcal; 0 point: no intake), fatty acids (10 points: (PUFA + MUFA)/SFA ≥2.5; 0 point: ≤1.2), refined grains (10 points: ≤1.8 oz eq./1000 kcal; 0 point: ≥4.3 oz eq./1000 kcal), sodium (10 points: ≤1.1 g/1000 kcal; 0 point: ≥2.0 g/1000 kcal), added sugar (10 points: ≤6.5% of energy; 0 point: ≥26% of energy), and saturated fats (10 points: ≤8% of energy; 0 point: ≥16% of energy) [29]. 

### 2.4. Covariates

For analysis of nutrients and food groups, age and energy intake were included as continuous variables in the analysis; whereas gender (male, female), race/ethnicity (Non-Hispanic White, Non-Hispanic Black, Mexican American, Other Hispanic, Other Race), and ratio of family income to poverty (≤1.85, 1.86–3.49, ≥3.50, with a higher value indicating higher income) were included as categorical variables in the analysis [30]. Another model without energy intake in the covariates was also assessed when comparing nutrient intakes between RTE cereal eaters and non-eaters. For analysis of HEI-2015 data, the same list of covariates was used, except that energy intake was not included as HEI-2015 scores were calculated based on intake per 1000 kcal [29]. The covariates were selected based on common sociodemographic characteristics that have been demonstrated to be plausibly associated with dietary intake and diet quality.

### 2.5. Sensitivity Analyses

Three sensitivity analyses were conducted for associations between RTE cereal consumption and nutrient intake in adults 18 years or older. First, education (high school graduate or less, some college education or associate degree, college graduate or above) and smoking status (current smokers, former smokers, non-smokers) were used as additional covariates to examine potential residual confounding. Second, 4318 participants in the study sample had two reliable dietary recalls; association between frequency of RTE cereal consumption (0 day, 1 day, 2 days) and nutrient intake, as estimated from average of two dietary recalls, was examined in this subgroup. Third, a stratified analysis by gender was conducted.

### 2.6. Statistical Analysis

SAS 9.3 (SAS Institute, Cary, NC, USA) was used for data analysis. Two-year sample weights and SAS survey procedures were applied to account for the NHANES survey design. For differences between RTE cereal eaters and non-eaters, categorical variables were compared by survey Chi-square tests, whereas continuous variables were compared by survey *t* tests or multivariable linear regression. Percentage below EAR in RTE eaters and non-eaters was compared by the Z-statistic [31]. Data were presented as weighted percentages or least squares means with standard errors. Bonferroni-corrected *p* values were applied to account for multiple tests in the two separate age groups. For example, as the nutrient and energy data included 24 outcomes, *p* < 0.05/48 = 0.001 was considered statistically significant for these outcomes. 

## 3. Results

### 3.1. Prevalence of RTE Cereal Consumption over Time

Figure 1 presents the prevalence of RTE cereal consumption in US adults across different NHANES cycles. In 2003–2004, 20% of US adults ≥18 years were RTE cereal eaters; it peaked at 24% in 2009–2010 but declined subsequently. In 2015–2016, the prevalence of RTE cereal consumption was 19% in the US adult population. Adults 55 years or older were more likely to be RTE cereal eaters than adults aged 18–54 years. Combining data from 2003–2004 to 2015–2016 revealed 27% of adults 55 years or older were RTE cereal eaters compared to 18% in adults aged 18–54 years. The number of RTE cereal eaters and non-eaters in different NHANES cycles as well as combined NHANES cycles can be found in Appendix A.

### 3.2. Demographics of Participants in NHANES 2015–2016

Table 1 presents the demographic characteristics of participants by RTE cereal consumption status. Average age was not different between RTE cereal eaters and non-eaters in adults aged 18–54 years or in adults 18 years or older; however, in adults aged 55 years or older, average age of RTE cereal eaters was higher than that in non-eaters (*p* = 0.003). There was no significant difference in gender distribution by RTE cereal consumption status in any of the age groups examined. RTE cereal consumption was not associated with race/ethnicity in adults aged 18–54 years; however, RTE cereal consumption was associated with race/ethnicity in adults aged 55 years or older, as well as in adults aged 18 years or older (*p* < 0.0001 for both) in that among both age groups a higher percentage of non-Hispanic white consumed RTE cereal. RTE cereal consumption was associated with family income to poverty ratio in 18–54 years old adults (*p* = 0.030); specifically, there was a higher percentage of RTE cereal eaters in the higher income category. However, the association was not found in adults 55 years or older, or in adults 18 years or older. No significant association between RTE cereal consumption and education was found (*p* > 0.05 for all); however, RTE cereal eaters were more likely to be non-smokers (*p* < 0.05 for all). In addition, RTE cereal eaters were more likely to consume breakfast compared to non-eaters in all age groups (*p* < 0.0001 for all).

### 3.3. Difference in Energy and Nutrient Intakes between RTE Cereal Eaters and Non-Eaters

Table 2 presents the percent difference in energy and nutrient intakes between RTE cereal eaters and non-eaters, adjusted for demographic characteristics (Model 1) and demographic characteristics plus energy intake (Model 2). Unadjusted results are reported in Appendix A.

There was no significant difference in energy intake between RTE cereal eaters and non-eaters when age groups were examined separately or pooled together (all *p* ≥ 0.001). When energy intake was included in the covariates, RTE cereal eaters aged 18 years or older had significantly higher intake of total carbohydrate, total sugar, dietary fiber, calcium, iron, magnesium, potassium, zinc, vitamin A, thiamin, riboflavin, niacin, vitamin B_6_, folate, vitamin B_12_, and vitamin D, and significantly lower intake of total fat (all *p* < 0.001), than non-eaters. Intake of added sugar, saturated fat, protein, sodium, vitamin C, and vitamin E did not significantly differ by RTE cereal consumption status (all *p* ≥ 0.001). Similar results were found when age groups were examined separately, except that intake of total sugar and vitamin B_12_ were not different by RTE cereal consumption status in adults 55 year or older (both *p* ≥ 0.001), and intake of potassium was not different by RTE cereal consumption status in adults 18–54 years (*p* = 0.002).

The results generally remained the same when only adjusted for demographic covariates, except that total fat intake was not different in any of the age groups examined (all *p* ≥ 0.001); intake of potassium was significantly higher for RTE cereal eaters in adults aged 18–54 years (*p* = 0.0006); and intake of vitamin C was significantly higher in RTE cereal eaters in adults aged 18 years or older (*p* = 0.0003), compared to non-eaters in the same age group.

Including education and smoking status as additional covariates in Model 2 did not materially change the associations between RTE cereal consumption and intake of energy and nutrients (Appendix A) with the exception that sodium intake was significantly associated with RTE cereal consumption with lower intake of sodium in RTE eaters than non-eaters (*p* = 0.0006); by contrast, the association between sodium intake and RTE cereal consumption was not statistically significant in Model 2.

Results on nutrient intake by frequency of RTE cereal consumption in participants with two reliable dietary recalls are presented in Appendix A. Very similar results on associations between RTE cereal consumption and nutrient intake were found as compared to Model 2 in the main analysis. The only difference was that intake of saturated fat was inversely and significantly associated with frequency of RTE cereal consumption using two-day dietary recalls (*p* = 0.0005), whereas it was not significantly associated with RTE cereal consumption using Day 1 dietary recall.

Stratification of the analysis by gender showed a similar pattern of results in men and women (data not shown).

### 3.4. Percentage of the Population below the Estimated Average Requirement for Nutrients

RTE cereal eaters were more likely to meet most nutrient intake recommendations than non-eaters (Table 3). No more than 1% of adult RTE cereal eaters aged 18 years or older were below the EAR for thiamin, riboflavin, niacin, folate, vitamin B_6_, vitamin B_12_, and iron. The percentages in non-eaters from the same age group, by contrast, were 11%, 5%, 3%, 21%, 19%, 8%, and 5%, respectively, which were significantly higher. Significant differences in the percent below EAR between RTE cereal eaters and non-eaters were also seen for calcium (27% vs. 50%), zinc (3% vs. 21%), vitamin A (9% vs. 56%), and vitamin D (83% vs. 98%). No significant difference in the percent below EAR for vitamin C (35% vs. 50%) was found. When adults 18–54 years and 55 years or older were examined separately, similar differences were found with a few exceptions: there was no significant difference by RTE cereal consumption status in percent below EAR for niacin and vitamin D in both age groups, and for vitamin B_12_ and iron in adults 55 years or older. 

### 3.5. Contribution of RTE Cereal to Daily Energy and Nutrients Intake in RTE Cereal Eaters and in the Total Population

Among RTE cereal eaters, RTE cereal contributed to 10% of energy intake in adults aged 18 years or older. The number was 2% when both RTE cereal eaters and non-eaters were combined as the total population. As shown in Figure 2, RTE cereal was a major food source of whole grains and several vitamins and minerals, such as folate, iron, vitamin B_6_, and vitamin B_12_. For example, RTE cereal contributed to 56% of daily intake of whole grains in adult RTE cereal eaters, and 18% of daily intake of whole grains in the total population. Conversely, RTE cereal was not a major food source of sodium and saturated fat as it contributed to 1% of sodium intake and less than 1% of saturated fat in the total adult population. Results were similar when examining adults 18–54 years and 55 years or older separately.

Lastly, due to the frequent co-consumption of RTE cereal with milk (89% of RTE cereal was co-consumed with milk in US adults, data not shown), percent contribution to daily intake among RTE cereal eaters from RTE cereal and co-consumed milk combined were calculated and the results are presented in Appendix A. As expected, including co-consumed milk increased percent contribution to daily intake for many nutrients in adults 18 years or older, such as vitamin D (from 21% to 50%), calcium (from 6% to 25%), vitamin B_12_ (from 38% to 50%), riboflavin (from 24% to 35%), vitamin A (from 27% to 37%), potassium (from 7% to 15%), saturated fat (from 2% to 9%), total sugar (from 12% to 19%), protein (from 6% to 13%), magnesium (from 13% to 19%), zinc (from 30% to 35%), and total fat (from 3% to 7%).

### 3.6. Food Group Intake in RTE Cereal Eaters and Non-Eaters 

Among adults aged 18 years or older, RTE cereal eaters had significantly higher intake of total dairy, fluid milk, total fruits, whole fruit, and whole grains, as well as lower intake of total protein foods, compared to non-eaters (*p* < 0.003 for all, Table 4). Intake of fruit juices, total vegetables, total grains, and refined grain did not differ by RTE cereal consumption status (*p* ≥ 0.003 for all). When adults aged 18–54 years and 55 years or older were examined separately, a similar pattern was found, except RTE cereal consumption was not associated with intake of total fruits and whole fruit in adults aged 18–54 years, and intake of whole fruit and total protein foods were not different in adults aged 55 years or older (*p* ≥ 0.003 for all).

### 3.7. Diet Quality in RTE Cereal Eaters and Non-Eaters 

Adult RTE cereal eaters had significantly better diet quality than non-eaters as reflected by the total HEI-2015 score (57.6 ± 0.9 vs. 50.1 ± 0.4, *p* < 0.0001), and the difference remained statistically significant in the age-stratified analysis (both *p* < 0.0001, Table 5). RTE cereal eaters also had significantly higher HEI sub-scores (meaning intakes closer to dietary recommendations) for total fruit, whole fruit, whole grains, dairy, and sodium than non-eaters (all *p* < 0.002). There were no differences in HEI sub-scores for total vegetables, greens and beans, total protein foods, seafood and plant protein, fatty acids, refined grains, saturated fat, and added sugar (all *p* ≥ 0.002). Similar results were found in adults aged 18–54 years and adults aged 55 years or older, except that the HEI sub-score for sodium did not differ by RTE cereal consumption status (both *p* ≥ 0.002).

## 4. Discussion

The present study found that US adults who reported consumption of RTE cereal had significantly higher intake of several vitamins and minerals that are typically under-consumed, as well as significantly higher intake of total fruits, dairy, and whole grains than non-eaters. RTE cereal eaters were more likely to meet nutrient recommendations and they had better diet quality, as defined by the HEI 2015, than non-eaters. These results are consistent with previous findings in other countries [6,7,8,9] and further support conclusions from a recent systematic review on the association between RTE cereal consumption and nutritional outcomes [32].

The prevalence of RTE cereal consumption was higher in adults aged 55 years or older than adults aged 18–54 years; moreover, among adults aged 55 years or older, those who consumed RTE cereal had a significantly older average age than those who did not consume RTE cereal. Despite the difference in prevalence of consumption, the study also found nutrients and food groups that were significantly different by RTE cereal consumption status were very similar in the two age groups examined, thus indicating the nutrition benefit of RTE cereal consumption was not age-specific.

It was found that RTE cereal eaters had higher intake of several nutrients such as iron, folate, vitamin B_6_ and vitamin B_12_, and they were less likely to consume below the recommended amount of these nutrients. More importantly, RTE cereal contributed to at least 38% of dietary intake of these nutrients in adult RTE cereal eaters alone, and at least 10% of the intake in all US adults. The contribution of RTE cereal to nutrient adequacy as well as its impact on public health should not be neglected. Recently, Papanikolaou and Fulgoni [33] assessed the contribution of grain foods, including RTE cereal, to nutrient intakes in US adults; they found that RTE cereal provided 5–17% of fiber, folate, zinc, and B vitamins, although it contributed less than 3% of total energy. Using NHANES 2007–2010 dietary data, Rehm and Drewnowski [34] conducted a modeling study by replacing all non-RTE cereal breakfast food with RTE cereal on a per calorie basis and found that the modeled RTE cereal breakfast resulted in significantly higher intake of iron, folate, and many other vitamins and minerals, compared to reported breakfast foods.

Similar to findings from previous studies [1,3,4,5,9], RTE cereal eaters had significantly higher intake of dietary fiber than non-eaters in this study. As one of the shortfall nutrients in the United States, dietary fiber intake should be encouraged. RTE cereal consumption was also associated with higher intake of whole grains. As a major food source of whole grains, RTE cereal contributed to 56% of whole grain intake in RTE cereal eaters alone and 18% in all US adults. The average intake of whole grains was 1.48 ounce equivalents in RTE cereal eaters and 0.74 ounce equivalents in non-eaters. In the US, the current recommendation for whole grains intake from the 2015–2020 Dietary Guidelines for Americans is to consume half of grain intake as whole grains, or three ounce equivalents per day [11]. As most US adults did not meet this dietary recommendation even with RTE cereal consumption, education and advocacy are still needed to encourage consumption of whole grain products, given the important role of whole grains in a healthy dietary pattern and disease prevention. A recent study identified low intake of whole grains as the second leading dietary risk factor for death globally [35].

It was noted that intake of added sugar, sodium, and saturated fats did not differ significantly between RTE cereal eaters and non-eaters in this study. In fact, the HEI-2015 sub-score for sodium was significantly higher in RTE cereal eaters than non-eaters in adults aged 18 years or older, indicating RTE cereal eaters had better compliance to dietary guidelines on sodium than non-eaters. RTE cereal appeared to be a minor food source for these nutrients to limit, as it only contributed to 3% of added sugar, 1% of sodium and less than 1% of saturated fats in the diets of all US adults. When looking at contributions to nutrients in RTE cereal eaters alone, RTE cereal provided 16% of added sugar, 7% of sodium and 2% of saturated fat. The finding that RTE cereal eaters had a similar level of daily intake of added sugar as non-eaters supports previous findings that RTE cereal consumption was not positively associated with intake of added sugar [3,36].

Previous studies have reported that RTE cereal consumption was associated with breakfast quality [37] and total diet quality [36]. The current study also found that consumption of RTE cereal was associated with better diet quality. The better diet quality was reflected by food group consumption including higher intake of dairy products, fruits, and whole grains in RTE cereal eaters. While a causal relationship cannot be inferred from this cross-sectional study, fortified nutrients in RTE cereal and co-consumed foods (such as milk) may have contributed to higher intake of vitamins and minerals, whole grains, and total dairy; in addition, RTE cereal eaters may be more health-conscious resulting in better dietary intake and diet quality.

Fortification is an effective strategy to achieve recommended intake of nutrients and improve public health. For example, folic acid fortification in cereal grain products has been implemented in the United States since 1998; a recent report estimated that it reduced live-born spina bifida cases by 767 annually [38]. Results from the present study showed that, while almost all RTE cereal eaters had sufficient intake of B vitamins and iron, 83% of them were still consuming below the EAR for vitamin D. The 2015–2020 Dietary Guidelines for Americans recommends fortified RTE cereal as strategy to increase vitamin D consumption. Food manufacturers should also consider these nutrients when creating or reformulating products to help consumers achieve nutrient recommendations. 

The present study focused on differences between RTE cereal eaters and non-eaters regardless of when RTE cereal was consumed and, as such, comparisons between RTE cereal breakfast eaters and non-RTE cereal breakfast eaters were not assessed. A previous study examined the type of breakfast in American adults and reported a breakfast pattern including RTE cereal and low-fat milk had lower daily intake of added sugar, saturated fat, and sodium, along with higher intake of shortfall nutrients than breakfast skipper [36]. In Canada, consumers of breakfast with RTE cereal had higher intake of calcium, thiamin, vitamin D, and iron than other breakfasts [6]. The present study found that 21% of RTE cereal consumers had RTE cereal at eating occasions other than breakfast, such as snacks, lunch, or dinner (data not shown). The reasons for choosing RTE cereal for non-breakfast occasions may be possibly due to convenience, nutrient content, taste, or cost; further research to understand factors that influence food choice is warranted.

There have been concerns about consumption of processed foods and its associations with unhealthy dietary nutrient intakes or chronic diseases [39]. Nonetheless, it should be noted that processed foods are a very broad category with varying products and nutrient profiles [40]. Due to the cross-sectional nature of the data, the present study did not comprehensively evaluate associations between RTE cereal consumption with health outcomes, although a descriptive analysis was conducted (Appendix A). RTE cereal eaters were more likely to have good or excellent general health and there was a lower percentage of overweight/obese participants in RTE cereal eaters than that in non-eaters. However, presence of self-reported diabetes and hypertension did not differ by RTE cereal consumption status in this study. Previous studies have shown that whole grains and cereal fibers have been associated with reduced risk of type 2 diabetes and total and cause-specific mortality [41,42,43,44]. A recent systematic review concluded that consumption of RTE cereal, especially of fiber-rich or whole grain RTE cereal, may have beneficial effects on hypertension and type 2 diabetes [32]. The systematic review did note that most studies on RTE cereal were funded by food manufacturers [32]. Among studies that examined RTE cereal and dietary intake in adults [1,2,3,4,5,6,7,8,9], only one study [1] was supported exclusively from government grants, although similar results were reported. Additional research supported by a variety of funding sources is encouraged.

This study has several strengths. It comprehensively evaluated differences in nutrient intake between RTE cereal eaters and non-eaters using a nationally representative sample from the adult population in the US, and it included intake of RTE cereal not only from breakfast but also from other eating occasions. While most RTE cereal is consumed at breakfast, including other eating occasions when defining RTE cereal eaters is important as the contribution of RTE cereal to daily nutrition intake is not specific to breakfast. Another strength of the study is the detailed dietary information that was collected using 24-h dietary recalls including brand-specific nutrient information for RTE cereal. Therefore, the results in this study represent the types of RTE cereals that are popularly consumed; however, future research on the impact of different types of RTE cereal on dietary and health outcomes in adults is warranted.

Despite these strengths, there are several limitations to this study that should be acknowledged. First, the present study using cross-sectional data focused on dietary intake as the main outcome of interests, and it did not extend the scope of the study to comprehensively examine health-related outcomes such as body weight status or biomarkers of health, although a descriptive analysis on general health was provided. Further studies, especially prospective cohort studies, should evaluate associations between RTE cereal consumption and health outcomes. Second, improved dietary quality or higher nutrient intake observed may be confounded by factors not examined in this study, for example, the type of breakfast consumed, habitual intake of RTE cereal, or other lifestyle factors. Third, dietary supplement data for NHANES 2015–2016 were not available at the time of the analysis, therefore contribution of nutrients from dietary supplements were not assessed in this study. Fourth, although the sensitivity analysis using two-day dietary data revealed very similar results, our classification of RTE cereal eaters using one-day dietary recall data did not account for habitual consumption of RTE cereal; hence, there was a potential misclassification. Future studies that use food frequency questionnaires should address the association between frequency of RTE cereal consumption and dietary intake over a longer time frame. Lastly, due to the observational nature of our data, causal relationships cannot be established; therefore, future research should test whether RTE cereal could be an interventional strategy to increase whole grain and nutrient intakes.

## 5. Conclusions

For the first time, the study showed that RTE cereal consumption, regardless of eating occasions, was associated with better nutrient intake and diet quality in US adults using the most recent one-day dietary data in NHANES 2015–2016. Findings from this study can be considered by policy makers and food manufacturers. RTE cereal was a major source of whole grains and several vitamins and minerals, and its consumption was associated with higher intake of dietary fiber, dairy products, and total fruits, all of which are an essential part of a healthy diet pattern.

## Figures and Tables

**Figure 1 nutrients-11-02952-f001:**
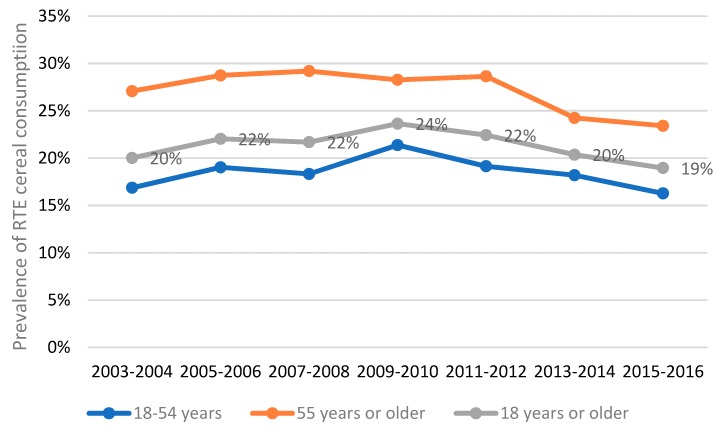
Prevalence of ready-to-eat (RTE) cereal consumption in US adults by age group in different cycles of the National Health and Nutrition Examination Survey using Day 1 dietary data.

**Figure 2 nutrients-11-02952-f002:**
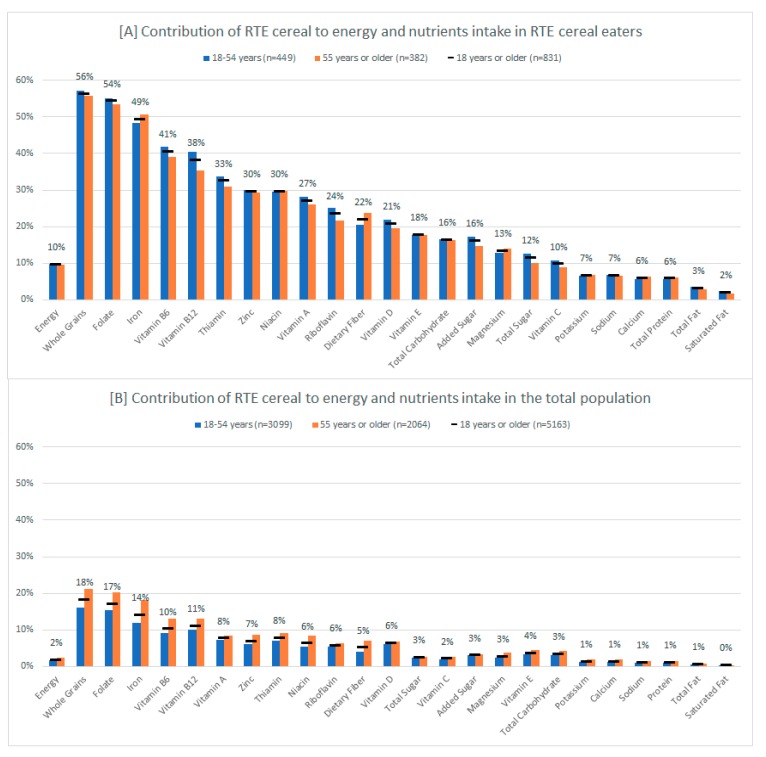
Contribution of ready-to-eat (RTE) cereal to daily intake of energy and nutrients in US adult RTE cereal eaters (**A**) and in US adults including both RTE cereal eaters and non-eaters (**B**) using Day 1 dietary data in National Health and Nutrition Examination Survey 2015–2016.

**Table 1 nutrients-11-02952-t001:** Demographic characteristics of participants by ready-to-eat (RTE) cereal consumption status in US adults ^1^.

	18–54 Years			≥55 Years			≥18 Years		
	RTE Cereal Eaters (*n* = 449)	RTE Cereal Non-Eaters (*n* = 2650)	*p* ^2^	RTE Cereal Eaters (*n* = 382)	RTE Cereal Non-Eaters (*n* = 1682)	*p* ^2^	RTE Cereal Eaters (*n* = 831)	RTE Cereal Non-Eaters (*n* = 4332)	*p* ^2^
Age (years)	37.3 ± 1.2	36.1 ± 0.2	0.30	68.4 ± 0.6	66.2 ± 0.4	**0.003**	51.7 ± 1.5	46.8 ± 0.7	0.11
Gender			0.37			0.74			0.57
Male	209 (47%)	1309 (51%)		195 (48%)	830 (47%)		404 (48%)	2139 (50%)	
Female	240 (53%)	1341 (49%)		187 (52%)	852 (53%)		427 (52%)	2193 (50%)	
Race/Ethnicity			0.06			**<0.0001**			**<0.0001**
Non-Hispanic White	154 (63%)	758 (56%)		206 (84%)	614 (73%)		360 (73%)	1372 (62%)	
Non-Hispanic Black	106 (13%)	593 (13%)		50 (5%)	357 (10%)		156 (9%)	950 (12%)	
Mexican American	82 (11%)	486 (12%)		51 (4%)	296 (5%)		133 (8%)	782 (10%)	
Other Hispanic	44 (5%)	343 (8%)		45 (2%)	246 (4%)		89 (4%)	589 (7%)	
Other Race	63 (8%)	470 (11%)		30 (6%)	169 (8%)		93 (7%)	639 (10%)	
Ratio of family income to poverty			**0.03**			0.91			0.15
≤1.85	220 (36%)	1357 (41%)		193 (37%)	920 (35%)		413 (36%)	2277 (39%)	
1.86–3.49	100 (18%)	610 (23%)		90 (23%)	372 (24%)		190 (20%)	982 (23%)	
≥3.50	129 (47%)	683 (36%)		99 (40%)	390 (40%)		228 (44%)	1073 (38%)	
Education ^3^			0.24			0.28			0.13
≤High school	181 (29%)	1187 (37%)		172 (35%)	877 (38%)		353 (32%)	2064 (37%)	
Associate degree	144 (36%)	802 (33%)		104 (29%)	463 (34%)		248 (33%)	1265 (33%)	
≥College graduate	124 (35%)	661 (31%)		106 (36%)	340 (28%)		230 (35%)	1001 (30%)	
Smoking status ^4^			**0.01**			**0.03**			**0.001**
Current smokers	62 (12%)	568 (21%)		39 (9%)	287 (17%)		101 (10%)	855 (20%)	
Former smokers	71 (22%)	371 (17%)		144 (38%)	585 (37%)		215 (29%)	956 (24%)	
Non-smokers	316 (67%)	1708 (61%)		199 (53%)	804 (46%)		515 (60%)	2512 (56%)	
Missing	0 (0%)	3 (0%)		0 (0%)	6 (0%)		0 (0%)	9 (0%)	
Breakfast consumption ^5^			**<0.0001**			**<0.0001**			**<0.0001**
Yes	413 (91%)	1919 (74%)		371 (98%)	1381 (82%)		784 (94%)	3300 (77%)	
No	36 (9%)	731 (26%)		11 (2%)	301 (18%)		47 (6%)	1032 (23%)	

^1^ Data are from the National Health and Nutrition Examination Survey 2015–2016. Adults 18 years and older with complete Day 1 24-h dietary recalls were included in the analysis. Data are presented as mean ± standard errors for continuous variables and n (weighted percent) for categorical variables. ^2^
*p* values for categorical variables are based on *Chi*-square test for surveys and *p* values for continuous variables are based on t-test for surveys; significant *p* values were shown in bold. ^3^ Two participants from RTE cereal non-eaters had missing information on education and were not included. ^4^ Smoking status was defined based on responses to questions “have you smoked at least 100 cigarettes in your entire life” and “do you now smoke cigarettes” in the Smoking-Cigarette Use Survey. ^5^ Breakfast consumption was defined by at least 50 kcal calorie intake from breakfast eating occasions in Day 1 dietary recall.

**Table 2 nutrients-11-02952-t002:** Percent difference in energy and nutrient intake between ready-to-eat (RTE) cereal eaters and non-eaters in US adults ^1^.

	18–54 Years				≥55 Years				≥18 Years			
	Model 1 ^2^		Model 2 ^3^		Model 1^2^		Model 2 ^3^		Model 1 ^2^		Model 2 ^3^	
	% Difference	*p* ^4^	% Difference	*p* ^4^	% Difference	*p* ^4^	% Difference	*p* ^4^	% Difference	*p* ^4^	% Difference	*p* ^4^
Energy	3.5	0.12	N/A	N/A	5.9	0.05	N/A	N/A	4.0	0.02	N/A	N/A
Total carbohydrate	16.3	**0.0004**	12.9	**0.0001**	19.2	**<0.0001**	12.9	**<0.0001**	16.9	**<0.0001**	13.1	**<0.0001**
Total sugar	24.3	**0.0008**	20.6	**0.0005**	21.9	0.003	14.4	0.005	22.5	**<0.0001**	18.2	**<0.0001**
Added sugar	10.4	0.20	6.3	0.35	−2.6	0.78	−10.8	0.17	4.4	0.37	−0.3	0.93
Dietary fiber	27.4	**<0.0001**	24.5	**<0.0001**	36.9	**<0.0001**	31.7	**<0.0001**	30.9	**<0.0001**	27.7	**<0.0001**
Total fat	−4.7	0.10	−8.5	**0.002**	−3.6	0.27	−9.9	**<0.0001**	−4.7	0.02	−8.9	**<0.0001**
Saturated fat	−1.3	0.69	−5.3	0.09	−0.3	0.94	−6.9	0.002	−1.3	0.59	−5.7	0.01
Protein	−0.8	0.72	−3.6	0.08	5.2	0.34	0.4	0.92	1.1	0.72	−2.1	0.32
Calcium	31.8	**<0.0001**	28.7	**<0.0001**	38.3	**<0.0001**	32.2	**<0.0001**	33.4	**<0.0001**	29.9	**<0.0001**
Iron	66.7	**<0.0001**	63.5	**<0.0001**	81.2	**<0.0001**	74.3	**<0.0001**	72	**<0.0001**	68.4	**<0.0001**
Magnesium	13.5	**0.0006**	10.9	**0.0002**	20.6	**0.0002**	15.5	**0.0004**	15.9	**<0.0001**	12.8	**0.0001**
Potassium	12.6	**0.0006**	9.8	0.002	21.3	**<0.0001**	16.3	**<0.0001**	15.7	**<0.0001**	12.6	**<0.0001**
Sodium	−6	0.02	−8.9	0.002	−0.3	0.95	−5	0.12	−4.2	0.14	−7.5	0.001
Zinc	33.9	**<0.0001**	30.8	**<0.0001**	38.4	**<0.0001**	32.6	**<0.0001**	34.6	**<0.0001**	31.1	**<0.0001**
Vitamin A	85.6	**<0.0001**	82.6	**<0.0001**	55.9	**0.0009**	49.7	**0.0003**	72.7	**<0.0001**	69.4	**<0.0001**
Thiamin	38.6	**<0.0001**	35.1	**<0.0001**	39	**<0.0001**	32.4	**<0.0001**	38.1	**<0.0001**	34.7	**<0.0001**
Riboflavin	40.4	**<0.0001**	37.2	**<0.0001**	40.2	**<0.0001**	33.9	**<0.0001**	39.3	**<0.0001**	35.4	**<0.0001**
Niacin	17.2	**0.0001**	14.3	**0.0002**	29.3	**<0.0001**	24.5	**<0.0001**	21.4	**<0.0001**	18.2	**<0.0001**
Vitamin B_6_	44.1	**<0.0001**	41	**<0.0001**	62.1	**<0.0001**	57	**<0.0001**	50.3	**<0.0001**	47.7	**<0.0001**
Folate	100.3	**<0.0001**	96.6	**<0.0001**	96.8	**<0.0001**	90.1	**<0.0001**	98.5	**<0.0001**	94.7	**<0.0001**
Vitamin B_12_	73.6	**<0.0001**	70.2	**<0.0001**	99.2	0.003	88.3	0.001	81.1	**<0.0001**	77	**<0.0001**
Vitamin C	15.3	0.08	12.9	0.10	33.6	0.001	30.9	0.003	23.9	**0.0003**	21.6	0.001
Vitamin D	100.8	**<0.0001**	97.3	**<0.0001**	76.5	**<0.0001**	68.1	**<0.0001**	90.6	**<0.0001**	86.3	**<0.0001**
Vitamin E	19.2	0.06	15.6	0.12	13.9	0.03	7.9	0.12	16.5	0.02	12.6	0.07

^1^ Data are from the National Health and Nutrition Examination Survey 2015–2016. Adults 18 years and older with complete Day 1 24-h dietary recalls were included in the analysis. Percent difference = (Mean intake in RTE cereal eaters−Mean intake in RTE cereal non-eaters)/Mean intake in RTE cereal non-eaters × 100. N/A: not available. ^2^ Model 1 adjusted for age, gender, race/ethnicity, and ratio of family income to poverty. ^3^ Model 2 adjusted for age, gender, race/ethnicity, ratio of family income to poverty, and energy intake. ^4^
*p* ≤ 0.001 (shown in bold) was considered to be statistically significant after adjusting for multiple comparisons considering 24 outcomes and two age groups (0.05/(24 × 2) = 0.001).

**Table 3 nutrients-11-02952-t003:** Percentages of ready-to-eat (RTE) cereal eaters and non-eaters below Estimated Average Requirement for nutrient intake ^1^.

	18–54 Years			55 Years or Older			18 Years or Older		
	RTE Cereal Eaters (*n* = 449)	RTE Cereal Non-Eaters (*n* = 2650)	*p* ^2^	RTE Cereal Eaters (*n* = 382)	RTE Cereal Non-Eaters (*n* = 1682)	*p* ^2^	RTE Cereal Eaters (*n* = 831)	RTE Cereal Non-Eaters (*n* = 4332)	*p* ^2^
Vitamin A	5 ± 3%	56 ± 3%	**<0.0001**	16 ± 4%	55 ± 4%	**<0.0001**	9 ± 4%	56 ± 0%	**0.002**
Thiamin	0 ± 0%	9 ± 2%	**0.0004**	1 ± 1%	15 ± 2%	**<0.0001**	0 ± 0%	11 ± 1%	**0.0001**
Riboflavin	0 ± 0%	5 ± 1%	**<0.0001**	0 ± 0%	4 ± 1%	**<0.0001**	0 ± 0%	5 ± 1%	**0.0002**
Niacin	0 ± 0%	2 ± 0%	0.003	0 ± 0%	5 ± 2%	0.007	0 ± 0%	3 ± 1%	**<0.0001**
Vitamin B_6_	0 ± 1%	11 ± 1%	**<0.0001**	2 ± 2%	34 ± 3%	**<0.0001**	1 ± 0%	19 ± 2%	**<0.0001**
Folate	0 ± 0%	17 ± 2%	**<0.0001**	0 ± 0%	29 ± 3%	**<0.0001**	0 ± 3%	21 ± 2%	**<0.0001**
Vitamin B_12_	0 ± 0%	7 ± 1%	**<0.0001**	0 ± 0%	10 ± 3%	0.004	0 ± 0%	8 ± 1%	**<0.0001**
Vitamin C	40 ± 5%	47 ± 3%	0.21	32 ± 7%	56 ± 3%	0.005	35 ± 4%	50 ± 2%	0.003
Vitamin D	78 ± 11%	98 ± 0%	0.08	84 ± 4%	98 ± 1%	0.005	83 ± 0%	98 ± 2%	**<0.0001**
Calcium	16 ± 4%	39 ± 3%	**0.0001**	41 ± 3%	70 ± 3%	**<0.0001**	27 ± 3%	50 ± 3%	**<0.0001**
Iron	0 ± 0%	8 ± 2%	**0.0004**	0 ± 0%	2 ± 1%	0.03	0 ± 1%	5 ± 2%	**<0.0001**
Zinc	1 ± 1%	17 ± 2%	**<0.0001**	8 ± 3%	29 ± 3%	**0.0002**	3 ± 1%	21 ± 1%	**<0.0001**

^1^ Data are from the National Health and Nutrition Examination Survey 2015–2016. Usual intake was calculated using the National Cancer Institute method with two-day dietary recalls, then compared with dietary reference intake to calculate percentages of participants below Estimated Average Requirement. Standard errors were calculated using the Balanced Repeated Replication Method. Percentage below EAR between RTE eaters and non-eaters was compared by the Z-statistic. ^2^
*p* ≤ 0.002 (shown in bold) was considered to be statistically significant after adjusting for multiple comparisons considering 12 outcomes and two age groups (0.05/(12 × 2) = 0.002).

**Table 4 nutrients-11-02952-t004:** Intake of food groups in ready-to-eat (RTE) cereal eaters and non-eaters in US adults ^1^.

	18–54 Years			≥55 Years			≥18 Years		
	RTE Cereal Eaters (*n* = 449)	RTE Cereal Non-Eaters (*n* = 2650)	*p* ^2^	RTE Cereal Eaters (*n* = 382)	RTE Cereal Non-Eaters (*n* = 1682)	*p* ^2^	RTE Cereal Eaters (*n* = 831)	RTE Cereal Non-Eaters (*n* = 4332)	*p* ^2^
Total dairy (cup eq.)	2.06 ± 0.16	1.34 ± 0.13	**<0.0001**	1.59 ± 0.13	1.02 ± 0.14	**<0.0001**	1.86 ± 0.11	1.20 ± 0.09	**<0.0001**
Fluid milk (cup eq.)	1.25 ± 0.09	0.47 ± 0.07	**<0.0001**	1.10 ± 0.11	0.50 ± 0.16	**<0.0001**	1.19 ± 0.04	0.48 ± 0.05	**<0.0001**
Total fruits (cup eq.)	1.26 ± 0.15	0.89 ± 0.10	0.01	1.51 ± 0.14	0.97 ± 0.09	**0.0003**	1.35 ± 0.12	0.90 ± 0.07	**0.0001**
Fruit juices (cup eq.)	0.31 ± 0.07	0.27 ± 0.05	0.38	0.33 ± 0.06	0.24 ± 0.03	0.04	0.32 ± 0.04	0.25 ± 0.04	0.03
Whole Fruit (cup eq.)	0.95 ± 0.14	0.61 ± 0.07	0.02	1.18 ± 0.15	0.73 ± 0.08	0.003	1.03 ± 0.11	0.65 ± 0.06	**0.0007**
Total vegetables ^3^ (cup eq.)	1.38 ± 0.10	1.56 ± 0.12	0.08	1.52 ± 0.15	1.53 ± 0.13	0.91	1.43 ± 0.11	1.55 ± 0.11	0.11
Total Protein Foods ^3^ (oz. eq.)	6.36 ± 0.29	7.56 ± 0.15	**0.0017**	5.81 ± 0.19	6.62 ± 0.14	0.006	6.14 ± 0.21	7.19 ± 0.11	**0.0008**
Total grains (oz. eq.)	7.38 ± 0.22	6.91 ± 0.10	0.09	6.54 ± 0.21	6.13 ± 0.12	0.10	7.03 ± 0.26	6.57 ± 0.19	0.03
Whole grains (oz. eq.)	1.40 ± 0.15	0.70 ± 0.10	**<0.0001**	1.60 ± 0.16	0.81 ± 0.15	**<0.0001**	1.48 ± 0.11	0.74 ± 0.11	**<0.0001**
Refined grains (oz. eq.)	5.97 ± 0.20	6.21 ± 0.11	0.27	4.94 ± 0.27	5.32 ± 0.14	0.25	5.55 ± 0.24	5.83 ± 0.21	0.19

^1^ Data are from the National Health and Nutrition Examination Survey 2015–2016 and Food Patterns Equivalent Database 2015–2016. Adults 18 years and older with complete Day 1 24-h dietary recalls were included in the analysis. ^2^ Adjusted for age, gender, race/ethnicity, and ratio of family income to poverty, and energy intake. *p* ≤ 0.003 (shown in bold) was considered to be statistically significant after adjusting for multiple comparisons considering 10 outcomes and two age groups (0.05/(10 × 2) = 0.003). ^3^ Legumes were included in the protein food group rather than the vegetables food group.

**Table 5 nutrients-11-02952-t005:** Healthy Eating Index-2015 total score and sub-scores in ready-to-eat (RTE) cereal eaters and non-eaters in US adults ^1^.

		18–54 Years			≥55 Years			≥18 Years		
	Maximum Score	RTE Cereal Eaters (*n* = 449)	RTE Cereal Non-Eaters (*n* = 2650)	*p* ^2^	RTE Cereal Eaters (*n* = 382)	RTE Cereal Non-Eaters (*n* = 1682)	*p* ^2^	RTE Cereal Eaters (*n* = 831)	RTE Cereal Non-Eaters (*n* = 4332)	*p* ^2^
Total vegetables	5	2.8 ± 0.1	3.1 ± 0.1	0.05	3.3 ± 0.1	3.3 ± 0.1	0.87	3.0 ± 0.1	3.2 ± 0.0	0.10
Greens and beans	5	1.9 ± 0.1	1.8 ± 0.1	0.40	2.1 ± 0.2	1.8 ± 0.1	0.15	1.9 ± 0.1	1.8 ± 0.1	0.21
Total fruits	5	2.6 ± 0.1	1.9 ± 0.1	**0.0001**	3.2 ± 0.2	2.3 ± 0.1	**<0.0001**	2.9 ± 0.1	2.0 ± 0.1	**<0.0001**
Whole fruit	5	2.7 ± 0.2	1.6 ± 0.1	**0.0001**	3.3 ± 0.2	2.5 ± 0.1	**0.001**	2.9 ± 0.1	2.1 ± 0.1	**<0.0001**
Whole grains	10	4.0 ± 0.2	1.9 ± 0.1	**<0.0001**	4.9 ± 0.3	2.5 ± 0.1	**<0.0001**	4.4 ± 0.1	2.1 ± 0.1	**<0.0001**
Dairy	10	6.4 ± 0.2	4.3 ± 0.1	**<0.0001**	5.9 ± 0.3	3.8 ± 0.1	**<0.0001**	6.2 ± 0.2	4.1 ± 0.1	**<0.0001**
Total protein food	5	4.0 ± 0.1	4.4 ± 0.0	0.02	4.3 ± 0.1	4.4 ± 0.1	0.52	4.1 ± 0.1	4.4 ± 0.0	0.05
Seafood and plant proteins	5	2.6 ± 0.1	2.3 ± 0.1	0.04	2.8 ± 0.1	2.6 ± 0.1	0.21	2.7 ± 0.1	2.4 ± 0.0	0.02
Fatty acids	10	4.5 ± 0.3	5.3 ± 0.1	0.03	5.0 ± 0.3	5.7 ± 0.2	0.02	4.7 ± 0.2	5.4 ± 0.1	0.004
Sodium	10	5.0 ± 0.3	3.9 ± 0.1	0.003	4.9 ± 0.2	4.1 ± 0.1	0.009	5.0 ± 0.2	4.0 ± 0.1	**0.0004**
Refined grains	10	6.2 ± 0.2	5.6 ± 0.1	0.04	6.5 ± 0.3	5.8 ± 0.2	0.05	6.4 ± 0.2	5.7 ± 0.1	0.02
Saturated fat	10	6.4 ± 0.3	5.8 ± 0.1	0.08	6.8 ± 0.3	6.1 ± 0.2	0.005	6.5 ± 0.2	5.9 ± 0.1	0.007
Added sugar	10	6.7 ± 0.3	6.9 ± 0.1	0.33	7.5 ± 0.3	7.4 ± 0.1	0.73	7.0 ± 0.2	7.1 ± 0.1	0.64
Total HEI-2015	100	55.9 ± 1.0	49.1 ± 0.5	**<0.0001**	60.6 ± 1.2	52.2 ± 0.7	**<0.0001**	57.6 ± 0.9	50.1 ± 0.4	**<0.0001**

^1^ Data are from the National Health and Nutrition Examination Survey 2015–2016. Adults 18 years and older with complete Day 1 24-h dietary recalls were included in the analysis. ^2.^ Adjusted for age, gender, race/ethnicity, and ratio of family income to poverty. *p* ≤ 0.002 (shown in bold) was considered to be statistically significant after adjusting for multiple comparisons considering 14 outcomes and two age groups (0.05/(14 × 2) = 0.002).

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
