# Peer review of "Association between Ready-to-Eat Cereal Consumption and Nutrient Intake, Nutritional Adequacy, and Diet Quality in Adults in the National Health and Nutrition Examination Survey 2015–2016"

_nutrients, 2019, doi:10.3390/nu11122952_

Round 1
Reviewer 1 Report
Thank you for your edits! This is an excellent study and this reviewer in particular appreciates your care in also including milk as part of the equation in the analysis.
There were only two typos that I found:
line 405: interests should be replaced with "interest"
line 433: NAHNES should be replaced with "NHANES"
I am also concerned that as they stand, tables 3 through 5 in the manuscript need attention in terms of their layout for clarity.
Table 3: Please add RTE to each of the Non-eaters columns for clarity
I do not know if these two changes need to come from the authors or the editors:
Table 4 I believe, would be best turned sideways on a full page because, as it currently is laid out, the first column wraps too much and makes the rest of the table hard to read.
Table 5. This table has the same problem as Table 4 and should be turned for clarity.
Author Response
Thank you for your edits! This is an excellent study and this reviewer in particular appreciates your care in also including milk as part of the equation in the analysis.
There were only two typos that I found:
line 405: interests should be replaced with "interest"
Response: suggested change was made in line 441.
line 433: NAHNES should be replaced with "NHANES"
Response: suggested change was made in line 472.
I am also concerned that as they stand, tables 3 through 5 in the manuscript need attention in terms of their layout for clarity.
Table 3: Please add RTE to each of the Non-eaters columns for clarity
Response: Thank you for the suggestion. We have reviewed the format and changed “non-eaters” to “RTE cereal non-eaters” in all tables.
I do not know if these two changes need to come from the authors or the editors:
Table 4 I believe, would be best turned sideways on a full page because, as it currently is laid out, the first column wraps too much and makes the rest of the table hard to read.
Table 5. This table has the same problem as Table 4 and should be turned for clarity.
Response: Thank you for bringing this to our attention. We currently have both tables in landscape format and will work with the journal editors to ensure that the tables remain legible.
Reviewer 2 Report
as pre my previous evaluation I don't find a scientific merit in this publication.
Author Response
as pre my previous evaluation I don't find a scientific merit in this publication.
Response: While we respect the reviewer’s perspective, we have provided the justification for this research in our introduction by outlining the need for updated data on the role of RTE cereal in Americans’ diets based on previous research demonstrating the important contribution ready-to-eat cereal makes to nutrient and whole grain intake. Furthermore, RTE cereal is recommended in the 2015-2020 Dietary Guidelines for Americans as an important source of whole grain and under-consumed nutrients; therefore, understanding the overall diet quality of those that consume RTE cereal is relevant (Lines 48-51). Our research has implications for food manufacturers (lines 397-402). Our results also show that RTE cereal is an important source of whole grains and nutrients of public health concern and its consumption is associated with higher diet quality thereby supporting the role of RTE cereal in a healthy dietary pattern (Lines 452-455).
Reviewer 3 Report
The paper is well written, but the authors should mark the innovative character of the research. This should be marked into the Conclusion together practical applications and future remarks. The linguistic revision of whole manuscript should be carried out. The Introduction should be greatly implemented and enlarged.The authors in Material and Methods should mark the importance of food source and update Food Composition Database, with particular emphasis with data inclusion of processed foods, food preparations and complex food matricies. In this regard appropriate references should be added such as:
Durazzo et al. 2019 Journal of Food Composition and Analysis; Marconi et al. 2018 Foods; Durazzo et al. 2018DOI:10.1016/B978-0-08-100596-5.21337-1,In book: Reference Module in Food Science)
Author Response
The paper is well written, but the authors should mark the innovative character of the research. This should be marked into the Conclusion together practical applications and future remarks.
Response: Thank you. We noted in our introduction that this research used the most recent nationally representative data to study the role of RTE cereal consumption at lines 52-54.
We added the following sentence to our discussion at lines 448-450 to note that this was only observational data and that future research is needed:
“Lastly, due to the observational nature of our data, causal relationships cannot be established; therefore, future research should test whether RTE cereal could be an interventional strategy to increase whole grain and nutrient intakes.”
Furthermore, we highlighted the practical application of this work at the following sections:
Lines 45-48: “Information about the current consumption status of RTE cereal and its association with dietary intake is critical as it may inform upcoming 2020-2025 Dietary Guidelines for Americans, nutrition policy, and product reformulation.”
Lines 397-402: “Results from the present study showed while almost all RTE cereal eaters had sufficient intake of B vitamins and iron, 83% of them were still consuming below the EAR for vitamin D. The 2015-2020 Dietary Guidelines for Americans recommends fortified RTE cereal as strategy to increase vitamin D consumption. Food manufacturers should also consider these nutrients when creating or reformulating products to help consumers achieve nutrient recommendations.”
The linguistic revision of whole manuscript should be carried out.
Response: Several revisions have been made to spelling and language throughout the manuscript in response to the reviewers’ comments which have been noted in red text.
The Introduction should be greatly implemented and enlarged.
Response: We added the following information to the introduction at lines 48-51:
“A recent study found that US children who reported consuming RTE cereal had higher diet quality [12]. This study sought to determine if the same association between RTE cereal consumption and diet quality was present among US adults.”
The authors in Material and Methods should mark the importance of food source and update Food Composition Database, with particular emphasis with data inclusion of processed foods, food preparations and complex food matricies. In this regard appropriate references should be added such as:
Durazzo et al. 2019 Journal of Food Composition and Analysis; Marconi et al. 2018 Foods; Durazzo et al. 2018 DOI:10.1016/B978-0-08-100596-5.21337-1,In book: Reference Module in Food Science)
Response: We appreciate that the reviewer mentioned the importance of food composition databases in dietary studies. As such, we cited the food composition database (FNDDS 2015-2016) used in the NHANES in our method section, together with details for the resources and methods used in FNDDS at lines 74-79.
“FNDDS includes the nutritional composition of all foods and beverages reported in NHANES including both those prepared at home or purchased [18]. The USDA National Nutrient Database for Standard Reference Release 28 (SR 28) is used to determine the nutritional value of foods and beverages in FNDDS; approximately one-third of foods and beverages in FNDDS are a direct match to items in SR 28 and a recipe calculation approach is used for the remaining foods [18].”
We also appreciate the suggested references provided by the reviewer; however, after review of the references, we noted these were not directly applicable to our study as the focus of this study was not the development of a food composition database nor was the food composition of Italian dishes included in our research. In their place, we have added the following reference #18 which provides more details on the food composition database used in this study:
Rhodes, D.G.; Morton, S.; Martin, C.L.; Adler, M.E.; Hymes, M.A.; Garceau, A.O.; Kovalchik, A.; Sattgast, L.H.; Steinfeldt, L.C.; Clemens, J.C.; LaComb, R.P.; Moshfegh, A.J. 2015-2016 Food and Nutrient Database for Dietary Studies Documentation. Availabe online: https://www.ars.usda.gov/ARSUserFiles/80400530/pdf/fndds/2015_2016_FNDDS_Doc.pdf (accessed on 11 October 2019).
Reviewer 4 Report
The manuscript used the NHANES survey data to show that ready-to-eat cereal consumers were probably having a better/healthier diet profile than non-consumers of the same food. The design and statistics were properly done. However, the main problem that I would say is that the difference between RTE cereal eaters and non-eaters must be more than purely from nutrition perspectives. Do they generally have healthy lifestyles, such as lower proportions of smokers/drinkers? Do they have a lower body mass index averagely? Do they have probably a longer time doing regular exercise? Do they also have different education levels/social-economic background? Although education level was used as one of the covariates in models conducted in supplementary table S3, an overall image/impression of those participants and how they might be different according to RTE cereal eating habits in the study is lacking here. Some participants/families in the US might not have the custom to buy or to eat RTE cereals at all.
Another major concern is about miss-classification of the eaters. Among two 24-h food recalls, only data from the first date was used to classify people into RTE cereal eaters or not. If we look at Supplementary table S4, only 362 reported having RTE cereals on both days, indicating that many non-eaters in the first diet recall survey probably simply happened to be not having RTE cereal in the previous day of their survey. How would this affect your findings? Is there a question that asks about the average frequency of having RTE over the past week or month, which will probably lower the chance of miss-classification of the participants. If we combine both 24-h food recalls, ie, identify eaters as who reported having RTE cereals for at least one day of the two surveys, would this reduce misclassification, and also attenuate the difference the authors found about the nutrients intake between eaters and non-eaters? And ironically, the authors said using “intake on a single day is not a robust estimate for usual intake” in line 108-109.
My impression is that energy intake, estimated average requirement, and food choices should be largely different by gender. Should there be stratification analyses by gender rather than by age? Why the authors chose to divide participants by age of 55 is un-justified and sometimes confusing as their reporting jumping between age-stratified results and total adults (>18 y.o).
Table 2 shows the percent difference in energy and nutrient intake between eaters and non-eaters. Surprisingly, the definition of this “percent difference” is not found in either the method section or results near the table if it is the main finding.
Some other specific comments are listed below:
Abstract:
Line 27-28: “diet quality” is kind of vague and unspecific. Do you mean more balanced with a wider variety of food choices? Or simply the same meaning as the sentence before this - “more like to meet nutrient recommendations”?
Introduction:
Line 36: “certain vitamins and minerals” might be expanded into more details.
Line 37: “focused on RTE cereal consumed at breakfast alone”. Do you mean previous studies only use breakfast to classify RTE cereal eaters and non-eaters? Or did they only assess the nutrients intake of RTE cereals consumed in the morning? I think the nutrient profile of the cereals would depend more on the manufacturers rather than the time when they were consumed.
Materials and methods:
How much is the response rate for this survey roughly for each cycle?
Line 69-70, how did NHANES determine the reliability of the dietary recall? Is there any under- or over-reporting concern compared with some other gold standard?
The 24 outcomes as the authors tried to evaluate afterward need to be listed and described more in 2.3.1
The healthy score that the authors applied might need to be explained a little bit more here, such as the range of the score, and how to evaluate “compliance” based on this score.
Line 129: does poverty:income ratio mean the higher the poorer or with higher income?
Results and discussion:
How different the number would be if RTE eaters were defined by breakfast RTE consumers only?
If your findings are similar to previous findings of RTE eaters have a higher intake of nutrients and better quality. What kind of new information does your study add to the literature?
If RTE cereals were consumed other than breakfast, when do Americans also have RTE cereals frequently? What do you think the main reasons would be if they choose to have RTE cereals rather than other foods during periods other than breakfast in a day?
Author Response
The manuscript used the NHANES survey data to show that ready-to-eat cereal consumers were probably having a better/healthier diet profile than non-consumers of the same food. The design and statistics were properly done. However, the main problem that I would say is that the difference between RTE cereal eaters and non-eaters must be more than purely from nutrition perspectives. Do they generally have healthy lifestyles, such as lower proportions of smokers/drinkers? Do they have a lower body mass index averagely? Do they have probably a longer time doing regular exercise? Do they also have different education levels/social-economic background? Although education level was used as one of the covariates in models conducted in supplementary table S3, an overall image/impression of those participants and how they might be different according to RTE cereal eating habits in the study is lacking here. Some participants/families in the US might not have the custom to buy or to eat RTE cereals at all.
Response: We agree with the reviewer that the difference between RTE cereal eaters and non-eaters may be more than just dietary differences. The role of confounding variables is inherent to all observational research such as nutritional epidemiological research studies. Our data does not imply any causal relationship, as stated in the limitations in the manuscript, lines 447-448. We also noted that future interventional research may help refine our understanding of the association between RTE cereal consumption and diet quality (lines 448-450).
However, we addressed the role of likely potential confounding variables by adjusting our analyses for total energy intake, age, gender, race/ethnicity, and ratio of family income to poverty. Our sensitivity analyses also considered educational attainment. The list of covariates was selected based on common sociodemographic characteristics that have been demonstrated to be plausibly associated with dietary intake and diet quality. Differences in these variables between RTE cereal eaters and non-eaters are available in Table 1.
We did not include lifestyle factors, such as smoking status, alcohol intake, or physical activity or health related measures such as body mass index among our covariates because the outcomes included in this study were dietary intake only, not health-related outcomes such as risk of cardiovascular disease. These health-related outcomes would be more prone to confounding by lifestyles or health related measures. This information has been added to our methods section at lines 168-172.
As suggested by the reviewer, we added information on the education status by RTE cereal consumption status to Table 1. We found RTE cereal consumption was not associated with education status, and this was added to the results section in lines 220-221.
Another major concern is about miss-classification of the eaters. Among two 24-h food recalls, only data from the first date was used to classify people into RTE cereal eaters or not. If we look at Supplementary table S4, only 362 reported having RTE cereals on both days, indicating that many non-eaters in the first diet recall survey probably simply happened to be not having RTE cereal in the previous day of their survey. How would this affect your findings? Is there a question that asks about the average frequency of having RTE over the past week or month, which will probably lower the chance of miss-classification of the participants. If we combine both 24-h food recalls, ie, identify eaters as who reported having RTE cereals for at least one day of the two surveys, would this reduce misclassification, and also attenuate the difference the authors found about the nutrients intake between eaters and non-eaters? And ironically, the authors said using “intake on a single day is not a robust estimate for usual intake” in line 108-109.
Response: Each dietary data collection method has strengths and weaknesses. We agree that 24 hr recalls, which provide detailed dietary intake data for a single day, do not provide a robust estimate for usual intake for an individual. However, they do provide an accurate estimate of population mean intake on a given day (reference #25). Therefore, individuals in this study are accurately classified as RTE cereal eaters for a single day and the study outcome of dietary quality is based on intake data from the same day. This justification can be found in our methods at line 93-95.
Below we provided further explanation for why day-1 only data were used when we reported mean intakes of nutrients in the main analysis.
Use of one-day dietary data for mean intake is a common approach in published studies using NHANES data. Here are examples of some recent publications (references #20-24). We included these citations to show readers that we are not unique in this methodological approach at line 93. NHANES dietary data are meant to represent the average intake for the US population on a given day. While one day of data may not represent each individual’s usual intake, the mean does accurately represent the average population intake; adding more days of intake will reduce the distribution around the mean but would not shift the mean (reference #25, line 95). Our results for nutrient intakes are reflective of nutrient intakes of RTE cereal eaters on the day they consumed cereal so that we can quantify contribution of RTE cereal to daily nutrient intake.
This is different than studying the association between RTE cereal consumption and a health outcome, in which, we agree, usual intake of RTE cereal over a longer time frame would be necessary to classify individuals as a habitual RTE cereal consumers to determine if habitual RTE cereal consumption was associated with a long-term health outcome which have long periods of development. An in-depth explanation of different dietary collection methods, their strengths and weaknesses as well as the use of habitual intake vs. single day of intake to classify individuals depending on the outcomes of interest is beyond the scope of our paper.
Furthermore, the study did include a sensitivity analysis using 2 days of dietary recall data to classify RTE cereal consumption status. As shown in Table S4 and stated in the results section (lines 249-253), the conclusion on the associations between RTE cereal consumption and nutrient intake remains the same for both methods (1-day vs 2-day).
My impression is that energy intake, estimated average requirement, and food choices should be largely different by gender. Should there be stratification analyses by gender rather than by age?
Response: We chose to stratify our results by age, rather than gender, as we noted a difference in RTE cereal consumption by age but not between men and women, as seen in Table 1. Additionally, we did include gender as a covariate in our analysis.
However, as suggested by the reviewer, we conducted a gender-stratified analysis on adjusted nutrient intake in adults aged 18 years or older to see if the results differed by gender. Our results showed that the association between RTE cereal consumption status and dietary intake was similar in men and women (please see table below). While absolute intake of nutrients may be different between men and women, the association between RTE cereal consumption and nutrient intakes did not differ by gender. We have noted our stratified analysis by gender in the methods section at line 181 and in the results at line 254; we chose not to include the results table in the manuscript due to the already lengthy results section and supplementary data file, particularly as we found no differences in results between men and women.
|
|
Men |
Women |
||||
|
|
RTE cereal non-eaters |
RTE cereal Eaters |
P |
RTE cereal non-eaters |
RTE cereal Eaters |
P |
|
Total carbohydrate (g) |
269.4±2.3 |
308.5±3.9 |
<.0001 |
211.0±1.4 |
235.5±4.3 |
<.0001 |
|
Total sugars (g) |
109.9±2.1 |
130.3±3.9 |
0.0002 |
88.0±0.9 |
103.7±3.6 |
0.0004 |
|
Added sugar (tsp. eq.) |
17.7±0.5 |
17.0±1.0 |
0.49 |
13.3±0.2 |
13.7±0.8 |
0.59 |
|
Dietary fiber (g) |
17.9±0.3 |
23.0±0.8 |
<.0001 |
15.7±0.3 |
19.9±0.7 |
<.0001 |
|
Total fat (g) |
93.4±0.6 |
83.8±2.0 |
0.0003 |
71.6±0.6 |
65.9±1.5 |
0.0010 |
|
Saturated fat (g) |
30.1±0.2 |
28.5±0.9 |
0.11 |
22.5±0.3 |
21.0±0.7 |
0.05 |
|
Protein (g) |
95.0±1.0 |
92.2±2.6 |
0.35 |
70.3±0.6 |
69.9±1.3 |
0.76 |
|
Calcium (mg) |
939.5±9.5 |
1269.4±35.7 |
<.0001 |
769.3±11.2 |
959.7±23.1 |
<.0001 |
|
Iron (mg) |
14.0±0.1 |
24.3±0.6 |
<.0001 |
11.0±0.1 |
18.0±0.4 |
<.0001 |
|
Magnesium (mg) |
324.1±6.1 |
367.2±4.6 |
0.0003 |
269.6±4.9 |
303.2±10.5 |
0.0041 |
|
Potassium (mg) |
2780.7±26.4 |
3191.5±57.5 |
<.0001 |
2286.2±32.2 |
2513.1±61.3 |
0.0015 |
|
Sodium (mg) |
4036.0±46.5 |
3783.9±92.2 |
0.04 |
3068.3±34.0 |
2786.2±72.6 |
0.0011 |
|
Zinc (mg) |
12.1±0.1 |
16.2±0.4 |
<.0001 |
8.9±0.1 |
11.4±0.3 |
<.0001 |
|
Vitamin A, RAE (mcg) |
547.7±20.7 |
1018.1±51.2 |
<.0001 |
523.2±17.6 |
799.5±66.4 |
0.0015 |
|
Thiamin (mg) |
1.7±0.0 |
2.3±0.1 |
<.0001 |
1.3±0.0 |
1.7±0.1 |
<.0001 |
|
Riboflavin (mg) |
2.2±0.1 |
3.0±0.1 |
<.0001 |
1.6±0.1 |
2.2±0.1 |
<.0001 |
|
Niacin (mg) |
29.7±0.6 |
35.2±0.7 |
<.0001 |
20.6±0.3 |
24.6±0.6 |
<.0001 |
|
Vitamin B6 (mg) |
2.3±0.1 |
3.4±0.2 |
<.0001 |
1.6±0.1 |
2.4±0.1 |
<.0001 |
|
Folate, DFE (mcg) |
487.7±7.5 |
987.1±30.1 |
<.0001 |
392.3±6.7 |
734.7±20.7 |
<.0001 |
|
Vitamin B12 (mcg) |
5.0±0.2 |
8.9±0.6 |
<.0001 |
3.3±0.1 |
6.0±0.2 |
<.0001 |
|
Vitamin C (mg) |
82.0±2.6 |
98.5±3.3 |
0.005 |
77.6±2.9 |
94.9±8.5 |
0.04 |
|
Vitamin D (mcg) |
4.3±0.2 |
8.6±0.3 |
<.0001 |
3.6±0.1 |
6.2±0.3 |
<.0001 |
|
Vitamin E (mg) |
9.0±0.2 |
9.8±0.5 |
0.16 |
8.2±0.2 |
9.5±1.0 |
0.15 |
Adjusted for age, race/ethnicity, ratio of family income to poverty, and energy intake.
Why the authors chose to divide participants by age of 55 is un-justified and sometimes confusing as their reporting jumping between age-stratified results and total adults (>18 y.o).
Response: We stratified our results by age due to differences in the prevalence of RTE cereal consumption (see Figure 1) in these two age groups of adults. We have now added this justification to our methods section at lines 89-91. There is no universal definition of “older adults”, but we chose the age of 55 which corresponds to the “baby boomer” generation in the US, born during the 2 decades after the World War II (1945-1965) [https://www.cnn.com/2013/11/06/us/baby-boomer-generation-fast-facts/index.html].
Table 2 shows the percent difference in energy and nutrient intake between eaters and non-eaters. Surprisingly, the definition of this “percent difference” is not found in either the method section or results near the table if it is the main finding.
Response: As suggested by the reviewer, we added the following information in the footnote of Table 2 as well as the methods section (lines 116-118) to show how this was calculated: Percent difference = [mean intake in RTE cereal eaters – mean intake in non-eaters]/ mean intake in non-eaters*100%.
Some other specific comments are listed below:
Abstract:
Line 27-28: “diet quality” is kind of vague and unspecific. Do you mean more balanced with a wider variety of food choices? Or simply the same meaning as the sentence before this - “more like to meet nutrient recommendations”?
Response: We defined diet quality according to the Healthy Eating Index 2015 which is an established method developed by the United States Department of Agriculture and the National Cancer Institute to measure diet quality and compliance with the U.S. Dietary Guidelines (https://epi.grants.cancer.gov/hei/). Please see section 2.3.5 “Healthy Eating Index 2015 for more information (lines 143 to 157). The following change was also made to clarify the definition of diet quality:
Lines 345-346: “RTE cereal eaters were more likely to meet nutrient recommendations and they had better diet quality, as defined by the HEI 2015, than non-eaters.”
Introduction:
Line 36: “certain vitamins and minerals” might be expanded into more details.
Response: This was changed to “certain vitamins and minerals such as thiamin, riboflavin, niacin, folate, vitamin D, iron, and zinc” in lines 36-37.
Line 37: “focused on RTE cereal consumed at breakfast alone”. Do you mean previous studies only use breakfast to classify RTE cereal eaters and non-eaters? Or did they only assess the nutrients intake of RTE cereals consumed in the morning? I think the nutrient profile of the cereals would depend more on the manufacturers rather than the time when they were consumed.
Response: In previous studies using 24-hour dietary recalls, only RTE cereal consumed at breakfast was included to classify participants into RTE cereal eaters or non-eaters. The nutrient intake in their analyses was assessed from daily dietary intake, not intake from breakfast alone. We have noted this in our introduction at lines 38-39:
“Nonetheless, most of the previous studies focused on RTE cereal consumed at breakfast alone and its association with daily dietary intake.”
RTE cereal is typically consumed at breakfast. Indeed, in the current study we found that approximately 80% of RTE cereal was consumed at breakfast, which could account for why most previous research focused on the breakfast eating occasion only. Nonetheless, we agree with the reviewer that the contribution of RTE cereal to daily intake would not be dependent on the time it is consumed and hence we included all RTE cereal consumption occasions in our study.
Materials and methods:
How much is the response rate for this survey roughly for each cycle?
Response: According to NHANES, the response rate for interview in NHANES 2015-2016 was 61.3%. This is now added to the manuscript in line 70.
Line 69-70, how did NHANES determine the reliability of the dietary recall? Is there any under- or over-reporting concern compared with some other gold standard?
Response: According to NHANES, dietary recall status was defined as reliable if it met the following criteria: the first 4 steps of the 5-step Automated Multiple Pass Method completed; and food/beverages consumed for each reported eating occasion identified. The following information was added in lines 83-86.
“According to NHANES 2015-2016, a reliable dietary recall status was defined as an individual completing the first 4-steps of the 5-step of the Automated Multiple Pass method and if foods/beverages consumed for each reported eating occasion were identified [13].”
The 24 outcomes as the authors tried to evaluate afterward need to be listed and described more in 2.3.1
Response: we added the list of nutrients in section 2.3.1. in lines 111-114.
The healthy score that the authors applied might need to be explained a little bit more here, such as the range of the score, and how to evaluate “compliance” based on this score.
Response: The following information was added in lines 144-157 to explain the scoring algorithm and range of the scores in HEI-2015.
“The HEI-2015 score is based on the sum of 13 components for a maximum of 100 points (highest diet quality) [29]. Points are given for each component according to intake per 1000 kcal or percent of energy. The maximum and minimum scores for each component are as follows: total fruits (5 points: ³ 0.8 cup eq./1000 kcal; 0 point: no intake), whole fruit (5 points: ³ 0.4 cup eq./1000 kcal; 0 point: no intake), total vegetables (5 points: ³ 1.1 cup eq./1000 kcal; 0 point: no intake), greens and beans (5 points: ³ 0.2 cup eq./1000 kcal; 0 point: no intake), whole grains (10 points: ³ 1.5 oz eq./1000 kcal; 0 point: no intake), dairy (10 points: ³ 1.3 cup eq./1000 kcal; 0 point: no intake), total protein foods (5 points: ³ 2.5 oz eq./1000 kcal; 0 point: no intake), seafood and plant proteins (5 points: ³ 0.8 cup eq./1000 kcal; 0 point: no intake), fatty acids (10 points: (PUFA+MUFA)/SFA ³ 2.5; 0 point: £ 1.2), refined grains (10 points: £ 1.8 oz eq./1000 kcal; 0 point: ³ 4.3 oz eq./1000 kcal), sodium (10 points: £1.1 g/1000 kcal; 0 point: ³2.0 g/1000 kcal), added sugar (10 points: £6.5% of energy; 0 point: ³26% of energy), and saturated fats (10 points: £8% of energy; 0 point: ³16% of energy) [29].”
Line 129: does poverty:income ratio mean the higher the poorer or with higher income?
Response: We changed the term to “Ratio of family income to poverty” throughout manuscript and table footnotes to avoid confusions. A higher score indicates higher income; this information has been added to the methods section at lines 161-162.
Results and discussion:
How different the number would be if RTE eaters were defined by breakfast RTE consumers only?
Response: We reported the number of breakfast eaters in each group in Table 1 but not the differences in intake from breakfast RTE cereal eaters only, as many previous studies have examined this question. Our study design was different from these studies but our conclusion generally supports previous results. Please refer to lines 334-336.
“These results are consistent with previous findings in other countries [6-9] and further supports conclusions from a recent systematic review on the association between RTE cereal consumption and nutritional outcomes [32].”
If your findings are similar to previous findings of RTE eaters have a higher intake of nutrients and better quality. What kind of new information does your study add to the literature?
Response: Previous studies in adults, as we cited in our introduction, were either 1) conducted outside of the United States; 2) used older data sets; 3) focused on the breakfast eating occasion only when using 24-h dietary recalls. Our study used the most recent, nationally representative U.S. dietary intake data as noted at lines 52 and 54 of the introduction. Shifts in the composition of RTE cereal, changes in dietary patterns, and changes in RTE cereal consumption patterns in the U.S. all could have plausibly led to different associations between RTE cereal consumption and diet quality; therefore, this study provides the most recent data among US adults which has important implications for nutrition policy and U.S. dietary guidance.
If RTE cereals were consumed other than breakfast, when do Americans also have RTE cereals frequently? What do you think the main reasons would be if they choose to have RTE cereals rather than other foods during periods other than breakfast in a day?
Response: We found that 79% of RTE cereal eaters consumed RTE cereal at breakfast, 6% at lunch, 11% as snacks, and 8% at dinner (the total exceeds 100% because some people may consume RTE cereal at multiple occasions in one day; data not reported in manuscript). There may be several possible reasons for people to choose RTE cereal for non-breakfast occasions, such as personal preference, tastes, cost, convenience, etc. However, the NHANES does not have any information on this from the participants, thus, it remains unknown. The following information was added at lines XXX.
“The present study found that 21% of RTE cereal consumers had RTE cereal at eating occasions other than breakfast, such as snacks, lunch, or dinner (data not shown). The reasons for choosing RTE cereal for non-breakfast occasions may be possibly due to convenience, nutrient content, taste, or cost; further research to understand factors that influence food choice is warr
Round 2
Reviewer 2 Report
My comments were conveyed directly to the editors
Author Response
Reviewer 2
My comments were conveyed directly to the editors.
Response: Thank you. We responded to comments from the editor in the previous section.
Reviewer 4 Report
Although the authors did provided some answers to my concerns, they were still not able to provide the details that will interest the readers. I did not ask you to compare and conclude that the health status/lifestyle are different between the so-called and untruely defined RTE eaters and non-eaters. It seemed like the authors were trying to not show the actual differences that should be shown. I am not convinced that factors such as BMI or smoking are not related with food intake (but they are, as people with higher BMI will need to consume more energy, etc.) so they can be deemed not necessary to be shown. Many variables could be associated with people choosing to eat or not to eat RTE on a certain day.
Because of the cross-sectional study design, there is no need to say there are any causal relationships between RTE eating and BMI, diabetes or hypertension, etc.. How can a study about nutrition ignoring the facts about the health of participants'? I believe that the NHANES have done health examinations of the participants and a study using simply the nutrients intake data would be considered as with lower significance which is also not helpful in terms of providing evidence to help understand the population in the US and for public health decision making.
If the objective of yours is simply to see the nutrients intake for those who happened to eat RTE on that day of the survey then maybe the terms of "RTE eaters" and "non-eaters" should be avoided, otherwise, it would be confusing and misleading for the readers. I would suggest emphasizing on the fact that this definition under the current study is based on 1-day recall, with a considerable potential bias of misclassification. To eliminate exaggeration/further extrapolation may be done by readers/journalists, it is needed to show sentences which reflect what was compared such as: comparing people who ate RTE to those who did not on a single day of the survey.
Another thing that the authors failed to explain is about the stratification analysis by the age of 55. The reason they provided is one of their results that they cut the population into relatively younger and older groups and say that there is a difference by age. It may be true, but if someone chooses a cut-off value of 50 years old, or maybe other cut-off values to separate people into more than 2 age groups, then they may have other different results. Did you find any interaction effect of age or nonlinear association between age and nutrients intake so that there must be a dichotomous division in age? I think if the authors remove the age stratification then the main results of different nutrients intake between RTE eaters and non-eaters will mostly remain and then there is space to enlarge Table 1 and add more public health relevant variables. Besides, the current version of the discussion is also largely unrelated to the age-stratified results.
Last but not least, reading the responses from a webpage is rather difficult without a connection to the internet. Please try to respond with separated and well-formatted documents.
Author Response
Reviewer 4
Although the authors did provided some answers to my concerns, they were still not able to provide the details that will interest the readers. I did not ask you to compare and conclude that the health status/lifestyle are different between the so-called and untruely defined RTE eaters and non-eaters. It seemed like the authors were trying to not show the actual differences that should be shown. I am not convinced that factors such as BMI or smoking are not related with food intake (but they are, as people with higher BMI will need to consume more energy, etc.) so they can be deemed not necessary to be shown. Many variables could be associated with people choosing to eat or not to eat RTE on a certain day.
Because of the cross-sectional study design, there is no need to say there are any causal relationships between RTE eating and BMI, diabetes or hypertension, etc.. How can a study about nutrition ignoring the facts about the health of participants'? I believe that the NHANES have done health examinations of the participants and a study using simply the nutrients intake data would be considered as with lower significance which is also not helpful in terms of providing evidence to help understand the population in the US and for public health decision making.
Response: We appreciate the comments from the reviewer. We designed the study using methods similar to many published studies using NHANES dietary data, before our data analysis. During the manuscript writing we accurately and objectively reported findings from our study and cited those references for our methods. We further conducted several sensitivity analyses as suggested by reviewers or editors. We agree health condition or lifestyle factors may affect dietary choices and there can be a lot of variables in health conditions and lifestyle factors. We also understand that different reviewers may have different opinions on the exact list of covariates to be used in the analysis. Per recommendation from the editor, here is what we did with regards to the health condition and lifestyle factors.
Included smoking status as another covariate in a sensitivity analysis; commented that not examining other lifestyle factors as a limitation. Conducted a descriptive analysis on general health conditions, overweight/obesity, presence of self-reported diabetes and hypertension, by RTE cereal consumption status.
The results for these new analyses can be found in Supplementary Table S3 and Table S6, and discussions on the results can be found in lines 244-247, 428-432, 453-461.
If the objective of yours is simply to see the nutrients intake for those who happened to eat RTE on that day of the survey then maybe the terms of "RTE eaters" and "non-eaters" should be avoided, otherwise, it would be confusing and misleading for the readers. I would suggest emphasizing on the fact that this definition under the current study is based on 1-day recall, with a considerable potential bias of misclassification. To eliminate exaggeration/further extrapolation may be done by readers/journalists, it is needed to show sentences which reflect what was compared such as: comparing people who ate RTE to those who did not on a single day of the survey.
Response: The use of terms “RTE eaters” and “non-eaters” and stating “using 1-day dietary data” hopefully can avoid potential confusions. We reviewed manuscript and ensured the term “using 1-day dietary data” or a similar phrase was always used, especially in the abstract and footnotes of tables for results, and our conclusions. We also revised our discussion to address the editor’s comment regarding use of 1-day dietary data, lines 463-468.
“Fourth, although the sensitivity analysis using 2-day dietary data revealed very similar results, our classification of RTE cereal eaters using 1-day dietary recall data did not account for habitual consumption of RTE cereal, hence, there is a potential misclassification; future studies that use food frequency questionnaires should address the association on frequency of RTE cereal and dietary intake over a longer time frame.”
Another thing that the authors failed to explain is about the stratification analysis by the age of 55. The reason they provided is one of their results that they cut the population into relatively younger and older groups and say that there is a difference by age. It may be true, but if someone chooses a cut-off value of 50 years old, or maybe other cut-off values to separate people into more than 2 age groups, then they may have other different results. Did you find any interaction effect of age or nonlinear association between age and nutrients intake so that there must be a dichotomous division in age? I think if the authors remove the age stratification then the main results of different nutrients intake between RTE eaters and non-eaters will mostly remain and then there is space to enlarge Table 1 and add more public health relevant variables. Besides, the current version of the discussion is also largely unrelated to the age-stratified results.
Response: We have reported results in the total adult population for readers who are interested in the total population. The difference in the consumption pattern in the two age groups reported in table 1 also supported our age-stratification analysis. Age-stratification was designed prior to data analysis, as we were interested in the consumption pattern of people at different life stages. The age cut-off value of 55 years was chosen based on the baby boomers in the United States. The baby boomer generation refers to those who were born in the two decades after the World War II. The youngest baby boomers (born in 1964) would be aged 55 years in 2019.
As suggested by the reviewer, we included a discussion on the age-stratified analysis in lines 348-354.
“The prevalence of RTE cereal consumption was higher in adults aged 55 years or older; moreover, those who consumed RTE cereal in this age group had a significant older average age than those who did not consume RTE cereal. Despite the difference in prevalence of consumption, the study also found nutrients and food groups that were significantly different by RTE cereal consumption status was very similar in the two age groups examined; thus indicating the nutrition benefit of RTE cereal consumption was not age-specific.”
Last but not least, reading the responses from a webpage is rather difficult without a connection to the internet. Please try to respond with separated and well-formatted documents.
Response: We uploaded a well-formatted PDF file to the system for our cover letter and responses to all comments from editors and reviewers, in addition to our responses provided in this online form.